# Effects of Efficient Expression of *Vitreoscilla* Hemoglobin on Production, Monosaccharide Composition, and Antioxidant Activity of Exopolysaccharides in *Ganoderma lucidum*

**DOI:** 10.3390/microorganisms9081551

**Published:** 2021-07-21

**Authors:** Zi-Xu Wang, Na Li, Jun-Wei Xu

**Affiliations:** 1Faculty of Life Science and Technology, Kunming University of Science and Technology, Kunming 650500, China; W1192176787@163.com; 2Faculty of Science, Kunming University of Science and Technology, Kunming 650500, China

**Keywords:** *Ganoderma*, *Vitreoscilla* hemoglobin, exopolysaccharide, production, monosaccharide composition, antioxidant activity

## Abstract

A *Vitreoscilla* hemoglobin (VHb) gene was efficiently expressed by the optimization of codons and intron addition in *G. lucidum*. Expression of the VHb gene was confirmed by genome PCR, quantitative real-time PCR and carbon monoxide (CO)-difference spectrum analysis in the transformant. The effects of the efficient expression of VHb gene on production, monosaccharide compostion, and antioxidant activity of *G. lucidum* exopolysaccharides were studied. The maximum production of exopolysaccharides in the VHb gene-bearing transformant was 1.63 g/L, which was 1.5-fold higher than expression in the wild-type strain. Efficient expression of the VHb gene did not change the monosaccharide composition or distribution of molecular weight, but it increased the mole percentage ratio of galactose and mannose in *G. lucidum* exopolysaccharide. Exopolysaccharides from the transformant had higher scavenging 2,2-diphenyl-1-picrylhydrazyl (DPPH) and hydroxyl (OH) radical capacity and reducing power than those from the wild-type strain. These results may be helpful for increasing production and application of exopolysaccharides produced by *G. lucidum* fermentation.

## 1. Introduction

*Ganoderma lucidum* is a medicinal mushroom that has been used in Asia for more than 2000 years to treat various diseases [1]. It is believed to preserve human vitality and to increase longevity [2]. Polysaccharides are active constituents of *G. lucidum* [3] that possess many kinds of bioactivities such as antioxidant, anti-aging, anti-inflammatory, immunomodulation, and antitumor bioactivities [4].

High production of *G. lucidum* exopolysaccharides is required to meet commercial demands. Many approaches have been developed to increase the production of *G. lucidum* exopolysaccharides. These include the optimization of media and fermentation conditions [5,6,7], addition of elicitor [8,9], development of new bioprocessing strategies [10,11], and genetic engineering of *G. lucidum* [12,13].

Hemoglobin from the bacterium *Vitreoscilla* sp. (VHb) is an oxygen binding protein [14]. The expression of the VHb gene increases the production of polysaccharides in *Bacillus subtilis* [15], *Sphingomonas elodea* [16], *Sphingomonas* sp. HT-1 [17], *Aureobasidium melanogenum* [18], and *Phellinus igniarius* [19]. In *G. lucidum*, expression of the native VHb gene also results in increased production of polysaccharides in submerged culture conditions [20]. We found that the presence of the glyceraldehyde-3-phosphate dehydrogenase gene (*gpd*) intron 1 intron and codon optimization are important for efficient expression of heterologous genes in *G. lucidum* [21]. However, effects of the efficient expression of the VHb gene by codon optimization and intron addition on exopolysaccharide production have not been studied in *G. lucidum*.

The bioactivities of polysaccharides are closely related to their chemical compositions and structural characteristics [22,23,24]. For example, Soltani et al. found that monosaccharide composition and molecular weight affect the bioactivities of polysaccharides extracted from *Cordyceps sinensis* [25]. In *G. lucidum*, culture medium and culture conditions can change the monosaccharide composition of exopolysaccharides and structural features [26,27,28]. However, the effects of expression of the VHb gene on monosaccharide composition, molecular weight, and antioxidant activity of exopolysaccharides have not been evaluated in *G. lucidum*.

In this study, we expressed the VHb gene by codon optimization and addition of the *gpd* intron 1 into *G. lucidum*. The effects of expression of the VHb gene on the production, monosaccharide composition and molecular weight of exopolysaccharides were evaluated in a submerged culture of *G. lucidum*. The antioxidant activities of exopolysaccharides produced by the wild-type (WT) and VHb gene-bearing strains were analyzed and compared. These results may be helpful for the hyper-production and application of *G. lucidum* exopolysaccharides.

## 2. Materials and Methods

### 2.1. Strains and Culture Conditions

Monokaryotic *G. lucidum* 5.616-1 strain [29] was grown in a complete yeast medium (CYM) containing 10 g/L maltose, 20 g/L glucose, 2 g/L tryptone, 2 g/L yeast extract, 0.5 g/L MgSO_4_, 4.6 g/L KH_2_PO_4_ and 10 g/L agar. *Escherichia coli* strain JM109 was used for cloning experiments and was cultivated in Luria–Bertani (LB) agar plates at 37 °C. The seed culture and liquid shaking fermentation of *G. lucidum* mycelia were conducted as described by Xu et al. [30,31].

### 2.2. Construction of Plasmid pJW-EXP-Intron-Opvhb

The *Nhe*I-*gpd* intron 1 (*intron*)—the codon-optimized VHb gene (*opvhb*)—*Sma*I sequence (Appendix A) was synthesized by Sangon Co., Ltd. (Shanghai, China). This sequence was ligated into plasmid pJW-EXP [32] that was digested with *Nhe*I and *Sma*I, yielding plasmid pJW-EXP-intro-opvhb.

### 2.3. Genetic Transformation of G. lucidum Protoplasts and Identification of Transformants

*G. lucidum* protoplasts were prepared after a 2.5 h digestion of 5 d old cultured mycelia with 2.5% lywallzyme (Guangdong Institute of Microbiology, Guangdong, China) in 0.6 M mannitol at 30 °C. The plasmid pJW-EXP-intro-opvhb was transformed into these protoplasts using a previously described method [33]. The obtained transformants were verified by genome PCR amplification of a 1.7 kb fragment containing partial *gpd* promoter and opvhb gene sequence using the primers gpd-F (5′-TCGTTCAAGCCTCTTCAGACATT-3′) and opvhb-R (5′-GCTCTATGTCTTGCCTTGTCTCG-3′). The VHb activity of the transformant was determined by CO-difference spectrum analysis as described by Xu et al. [20]. VHb concentration was calculated using the extinction coefficient E_419–436_ nm = 274 mM^−1^ cm^−1^, as previously described [34].

### 2.4. Transcription Analysis of OPVHb Gene Using Quantitative Real-Time PCR (qRT-PCR)

Total RNA of transformants was extracted with a fungal Total RNA Isolation Kit (Sangon Co., Ltd., Shanghai, China), and then reverse transcribed to cDNA using the PrimeScript RT reagent Kit (Takara, Dalian, China) following manufacturer’s instructions. The transcription levels of the OPVHb gene in the transformant were measured by qRT-PCR assay according to a procedure described previously [12,13]. The *G. lucidum* 18S rRNA gene was used as a housekeeping gene [20]. The expression level of the OPVHb gene on day 1 is defined as 1.0, and expression levels at different cultivation times were expressed as the fold increase over the reference sample. The primers qRT-opvhb-F (5′-CGCCCACTACCCCATCGT-3′) and qRT-opvhb-R (5′-CGGCCTCGACCTGGATAAA-3′) were used for qRT-PCR analysis.

### 2.5. Production of Biomass and Exopolysaccharides

*G. lucidum* mycelia biomass was measured by centrifuging the collected culture at 10,000× *g* for 10 min, washing the precipitated mycelia three times with distilled water and drying at 50 °C. For determination of exopolysaccharide yield, the fermentation broth supernatant was combined with four volumes of 95% (*v*/*v*) cold ethanol overnight to precipitate the crude exopolysacharides. The precipitated exopolysaccharides were collected by centifugation at 13,000× *g* for 10 min and suspended in 1 M NaOH at 60 °C for 1 h. Then, the crude exopolysaccharide yield was determined by a phenol-sulfuric acid assay method described previously [12,13].

### 2.6. Extraction of G. lucidum Exopolysaccharides

The exopolysaccharides precipitated by cold ethanol were dissolved in distilled water and deproteinized by the method of Sevag et al. [35]. Then, the deproteinized exopolysaccharides solution was concentrated by vacuum evaporation, dialyzed with distilled water, and lyophilized for further study.

### 2.7. Determination of Exopolysaccharide Molecular Weight

The molecular weight was determined by high-performance gel permeation chromatography (HPGPC) using a Shimadzu LC-10A with a BRT105-104-102 column and refractive index detector (Shimadzu, Kyoto, Japan). The polysaccharide samples and dextran standards (5 kDa, 11.6 kDa, 23.8 kDa, 48.6 kDa, 80.9 kDa, 148 kDa, 273 kDa, 409.8 kDa, 667.8 kDa) were analyzed by a BRT105-104-102 column, and eluted with 0.05 M NaCl at a flow rate of 0.6 mL/min at 40 °C. The molecular weight was calculated by referencing the calibration curve.

### 2.8. Analysis of Monosaccharide Compositions

Monosaccharide compostions were analyzed according to the method of pre-column derivatization RP-HPLC [36]. A sample (5.0 mg) was hydrolyzed to monosaccharides with 4M trifluoroacetic acid (TFA) for 6 h at 120 °C. The hrydrolyzed products or monosaccharide standards were derivatized with 0.5 M 1-phenyl-3-methyl-5-pyrazolone (PMP) solution under alkaline condition (70 °C, 100 min). The derivatives were extracted by chloroform and analyzed by HPLC (1200, Agilent, Waldbronn, Germany). The HPLC conditions were as follows: column, Zorbax SB-C18 (4.6 mm × 250 mm); mobile phase, 0.1 M phosphate buffered solution (pH 6.7): acetonitrile (83:17, *v*/*v*); flow rate, 1 mL/min; detector, 245 nm; oven temperature, 30 °C.

### 2.9. Measurements of Antioxidant Activity

The reducing power was determined by the method of Ge et al. with some modifications [37]. Briefly, 0.25 mL of aqueous exopolysaccharide solution at various contentrations (1–6 mg/mL) was mixed with 0.25 mL of phosphate buffer (0.2 M, pH 6.6) and 0.25 mL potassium ferricyanide (1%, *w*/*v*) at 50 °C for 20 min. After adding 0.25 mL of 10% trichloroacetic acid, the mixture system was centrifuged at 3000× *g* for 10 min. Next, 0.5 mL supernatant was collected and mixed with 0.5 mL ferric chloride (0.1%, *w*/*v*). After incubation for 10 min, the absorbance against a blank (distilled water) was measured using a spectrophotometer at 700 nm. Increased absorbance indicated increased reducing power.

The 2,2-Diphenyl-1-picrylhydrazyl (DPPH) radical-scavenging activity was determined according to a previously reported method with some modification [37]. Briefly, a 0.5 mL exopolysaccharide sample was added to 0.5 mL DPPH solution (0.1 mmol/L) and allowed to react in darkness at room temperature for 30 min. After the mixture was centrifuged at 5000× *g* for 10 min, the absorbance of the supernatant was measured at 517 nm. The DPPH radical scavenging activity was calculated as follows:DPPH scavenging activity (%) = [1 − (A_1_ − A_2_)/A_3_] × 100%
where A_1_ is the absorbance of the first mixture (0.5 mL of sample solution and 0.5 mL of DPPH ethanolic solution), A_2_ is the absorbance of the second mixture (0.5 mL of sample solution and 0.5 mL of absolute ethanol), and A3 is the absorbance of the third mixture (0.5 mL of distilled water and 0.5 mL of DPPH ethanolic solution).

The hydroxyl radical-scavenging activity was determined according to the Fenton method with some modifications [38]. The reaction mixture that contained 0.5 mL of ferrous sulfate (6 mmol/L), 0.5 mL of salicylic acid ethanolic solution (5 mmol/L), 0.5 mL of sample solution, and 0.5 mL of hydrogen peroxide (8.8 mmol/L) was allowed to react at 37 °C for 30 min. After the mixture was centrifuged at 5000× *g* for 10 min, the absorbance of the supernatant was measured at 510 nm. The hydroxyl radical-scavenging activity was calculated as follows:hydroxyl radical-scavenging activity (%) = [1 − (A_1_ − A_2_)/A_3_] × 100%,
where A_1_ is the absorbance of the reaction systems with various concentrations of exopolysaccharides, A_2_ is the absorbance of a reaction system where salicylic acid was replaced with 0.5 mL of absolute ethanol, and A3 is the absorbance of a reaction system where the polysaccharide sample was replaced with 0.5 mL of distilled water.

### 2.10. Statistical Analysis

All fermentation experiments were carried out in triplicate, and all data are presented as the the mean ± standard deviation. Statistical significance (*p* < 0.05) was analyzed by two-tailed ANOVA.

## 3. Results and Discussion

### 3.1. Expression of the VHb Gene in G. lucidum

A previous study demonstrated that the presence of the *gpd* intron 1 in heterologous genes can increase levels of mRNA accumulation and protein expression in *G. lucidum* [21]. In contrast to a previous study [20], we constructed the plasmid pJW-EXP-intron-opvhb (Figure 1A) where the *gpd* intron 1-codon-optimized VHb gene was cloned in the plasmid pJW-EXP [32]. The codon-optimized VHb gene was regulated by the *gpd* promoter and the succinate dehydrogenase gene (*sdh*) terminator of *G. lucidum*. Consequently, the plasmid pJW-EXP-intron-opvhb was transformed into protoplasts of *G. lucidum*. We obtained *G. lucidum* transformants OPVHB (Figure 1B) after selection on CYM plates with carboxine. The proper integration of the *opvhb* into the transformant OPVHB was verified by genome PCR. Figure 1C shows a clear band for the fused *gpd* promoter and *opvhb* (−1.7 kb) fragment in the strain OPVHB, but no corresponding band was amplified in the WT strain. The biological activity of VHb in the transformant OPVHB was confirmed by carbon monoxide (CO) difference spectrum assays (Figure 1D). A maximal characteristic absorption at 418 nm, derived from absorbance from CO binding to VHb, was observed in crude extract of the transformant OPVHB, while no such absorbance was detected with the crude extract of the WT strain. These results illustrate that the transformant OPVHB expressed the biologically active VHb.

### 3.2. Profiles of VHb Expression in the Transformant OPVHB

The time courses of *opvhb* transcription levels were examined by qRT-PCR in the transformant OPVHB (Figure 2A). The *opvhb* transcription level reached its highest level on day 9. It then declined but maintained a higher value at day 12 than at the beginning of the study. The transcription level of *opvhb* on day 9 was 4.7 times the value at the beginning. The VHb contents of the transformant OPVHB at different cultivation times paralleled those observed for *opvhb* mRNA (Figure 2B). The maximim VHb content was at day 9. After day 9, there was a decrease in VHb until the end of fermentation. The content of VHb for the transformant OPVHB was about 40 nmol/g dry cell weight (DCW), and this value was about two-fold higher than that (7–20 nmol/g DCW) reported in *G. lucidum* bearing the native *vgb* [20]. This result is consistent with our previous study showing that the presence of a *gpd* intron 1 increased the expression level of heterologous green fluorescent protein gene in *Ganoderma* [21]. These results showed that *opvhb* was efficiently expressed in the transformant OPVHB.

### 3.3. Increased Production of Exopolysaccharides by Expression of Opvhb in G. lucidum

The time profiles of cell growth and production of exopolysaccharides were analyzed in the WT and OPVHB strains under liquid shaking culture condition. Figure 3A shows that the OPVHB strain carrying *opvhb* produced greater biomass (9.16 g/L) than the WT strains (8.12 g/L), although the cell growth patterns in both strains showed a similar trend. These results demonstrated that the active VHb in the engineered strain improved cell growth in liquid shaking culture. Similar observations were reported for *Aspergillus sojae* [39] and *Phellinus igniarius* [19], where increased cell growth was associated with VHb gene expression. Figure 3B shows that the production of exopolysaccharides increased until day 9 and then remained approximately constant thereafter in both the WT and OPVHB strains. The maximum production of exopolysaccharides in the OPVHB strain was 1.63 g/L, which was 1.5-fold higher than that in the WT strain. Additionally, the exopolysaccharide production in the OPVHB strain was 1.95-fold higher than that reported for *G. lucidum* bearing the native VHb gene [20]. Therefore, the expression of *opvhb* is an efficient strategy to increase exopolysaccharide production in a submerged culture of *G. lucidum*. Similarly, heterologous expression of the VHb gene increased production of gellan gum and exopolysaccharides in *Sphingomonas sp*. HT-1 and *Phellinus igniarium*, respectively [17,19]. The efficient expression of *opvhb* in the engineered strain may enhance the utilization of oxygen and facilitate energy generation. This may be beneficial to exopolysaccharide biosynthesis from simple sugars [14,18,40]. The engineered OPVHB strain can be used for the hyper-production of *G. lucidum* exopolysaccharides.

### 3.4. Effects of Opvhb Expression on Molecular Weight Distribution and Monosaccharide Composition of G. lucidum Exopolysaccharides

We examined effects of *opvhb* expression on the molecular weight (Mw) distribution of exopolysaccharides obtained from the WT and OPVHB strains under liquid shaking culture conditions. The Mw distribution profiles showed that the crude exopolysaccharides from the WT strain exhibited two dominant peaks accompanied by two minor components (Figure 4). As calculated by the standard curve, the weight-average Mws of the dominant peaks were 11.5 and 15.4 × 10^3^ kDa. The Mw distribution of the exopolysaccharides from the OPVHB strain was the same as that of the WT, indicating that *opvhb* expression did not change the distribution of the Mw. To investigate the effect of *opvhb* expression on monosaccharide compositions of crude exopolysaccharides, exopolysaccharides from the WT and OPVHB strains were analyzed using the HPLC. Table 1 shows that both exopolysaccharides are composed of glucose, galactose, mannose, rhamnose, fucose, glucuronic acid, arabinose and xylose. The main monosaccharides were glucose, mannose and galactose. This is consistent with a previous report showing that *G. lucidum* exopolysaccharides contain seven monosaccharides, predominantly glucose, galactose, mannose, rhamnose, fucose, arabinose and xylose [27]. The molar ratios of glucose, galactose, mannose, rhamnose, fucose, glucuronic acid, arabinose, and xylose were 36.3: 19.7: 24.9: 2.6: 4.3: 6.5: 2.9: 2.8 for the exopolysaccharides from WT strain, whereas the ratios were 35.4: 23.1: 31.5: 1.4: 3.2: 3.4: 1.1: 0.9 for the exopolysaccharides from the OPVHB strain. Expression of *opvhb* in *G. lucidum* increased the ratio of galactose and mannose, but decreased the ratio of rhamnose, fucose, glucuronic acid, arabinose, and xylose in exopolysaccharides. These results indicate that *opvhb* expression can be an effective strategy for changing the ratio of monosaccharides in *G. lucidum* exopolysaccharides. Previous work showed that the presence of the VHb gene regulated the transcription of polysaccharide biosynthetic genes in *G. lucidum* [20]. The *opvhb* expression might affect the exopolysaccharide biosynthetic pathway and induce sugar metabolic shifts. Improved understanding of the underlying mechanisms will be possible when the polysaccharide biosynthetic pathway is better understood.

### 3.5. Effect of Efficient Opvhb Expression on Antioxidant Activity of Exopolysaccharides

Hydroxyl free radicals are reactive oxygen species that are most toxic to organisms. Therefore, we investigated the scavenging capacity of hydroxyl radicals by exopolysaccharides from the WT and OPVHB strains. Figure 5A shows that the scavenging capacity of exopolysaccharides is positively correlated with the dose.

The maximum scavenging rate of exopolysaccharides from the OPVHB strain was 58.1 ± 4.0% when the concentration was 2.5 mg/mL. This was 1.94-fold higher than that of the WT strain. DPPH is another stable free radical that was used for quantifying scavenging activity of free radicals. Figure 5B shows the inhibitory effect of both exopolysaccharides on DPPH. The scavenging effect of both samples was dose dependent. The exopolysaccharides from the OPVHB strain showed a more effective scavenging capacity than those from the WT strain in the dose range of 1–5 mg/mL. The exopolysaccharides from the OPVHB strain exhibited the highest DPPH scavenging rate of 75.87% when the dose was 5 mg/mL. However, the clearance rate of the exopolysaccharides from the WT strain at 5 mg/mL was only 60.92%. The total reducing power was associated with the antioxidant activity. The total reducing power of exopolysaccharides displayed concentration-dependent scavenging in the range of 1–5 mg/mL (Figure 5C). The maximum reducing powers of exopolysaccharides from the OPVHB and WT strains were 0.49 and 0.43, respectively, indicating the reducing power of exopolysaccharides from OPVHB was slightly stronger than that from the WT strain. The antioxidant activity of exopolysaccharides depends mainly on physiochemical and structural characteristics, such as molecular weight, contents of monosaccharide constituent and hexuronic acid and glycosidic linkages [41,42]. The good antioxidant activity of exopolysaccharides from the OPVHB strain may be related to their structural characteristics.

## 4. Conclusions

VHb was efficiently expressed in *G. lucidum* by addition of the *gpd* intron 1 at 5′ upstream of the VHb gene. Efficient expression of the VHb gene led to increased production of exopolysaccharides in submerged culture of *G. lucidum*. The ratio of galactose and mannose in exopolysaccharides and the antioxidant activity of exopolysaccharides were greatly enhanced by efficient expression of the VHb gene in *G. lucidum*. This study may be beneficial for the hyper-production of *G. lucidum* exopolysaccharides in large-scale fermentation. The information may aid further study of the relationship between exopolysaccharide structure and its function.

## Figures and Tables

**Figure 1 microorganisms-09-01551-f001:**
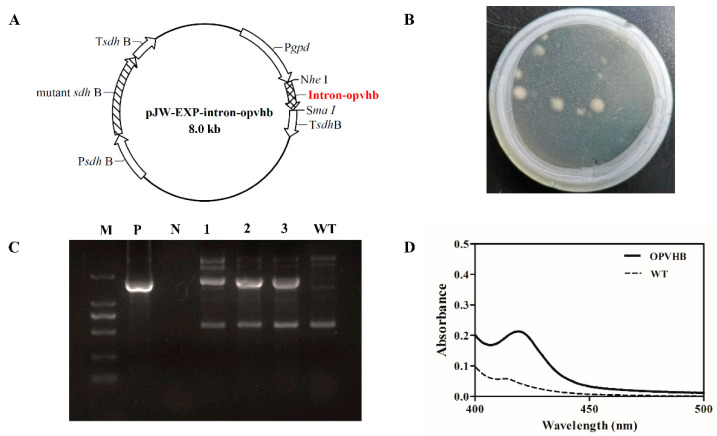
(**A**) Structure of the plasmid pJW-EXP-intron-opvhb. (**B**) Selection of OPVHB transformants on a CYM plate with 2 mg/L carboxine. (**C**) Identification and characterization of the transformants OPVHB by genome PCR. Lane M, DNA marker DL2000; Lane P, pJW-EXP-intron-opvhb as positive control; Lane N, negative control; Lane WT, wild-type strain; Lane 1, 2, 3, the transformants OPVHB. (**D**) CO-difference spectra analysis of the WT and OPVHB strains.

**Figure 2 microorganisms-09-01551-f002:**
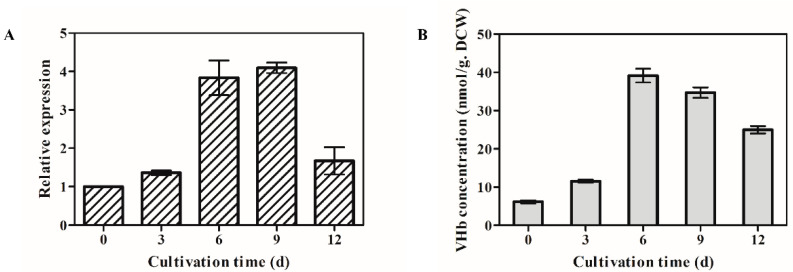
Transcriptional levels of the VHb gene (**A**) and VHb concentrations (**B**) in the submerged cultured OPVHB strain. The expression level of the *opvgb* on day 1 is defined as 1.0, and expression levels at different cultivation times were expressed as the fold increase over the reference sample. The VHb concentration was calculated using the extinction coefficient E_419–436_ nm = 274 mM^−1^ cm^−1^. d, days.

**Figure 3 microorganisms-09-01551-f003:**
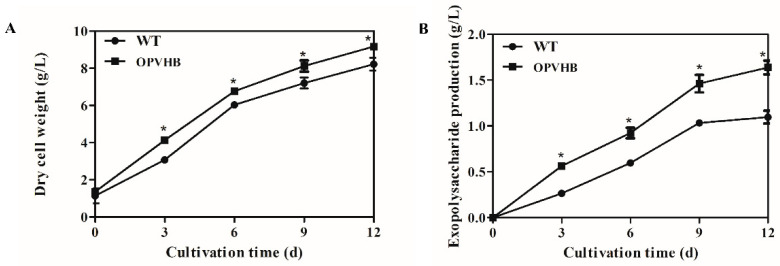
Kinetic profiles of cell growth (**A**) and production of exopolysaccharides (**B**) in the WT and OPVHB strains. * indicates significantly different from value for WT (*p* < 0.05).

**Figure 4 microorganisms-09-01551-f004:**
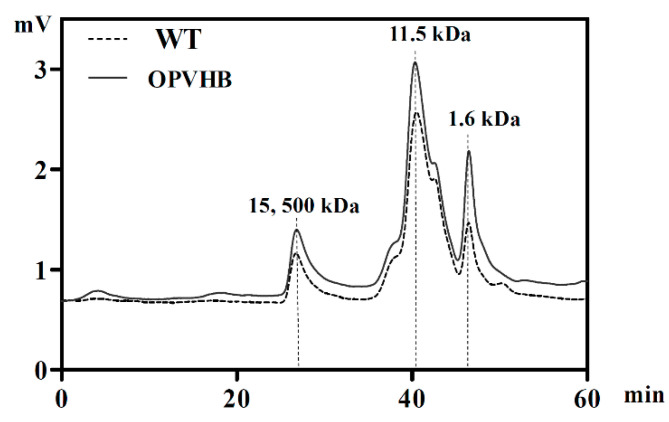
High-performance gel permeation chromatography of exopolysaccharides from the WT and OPVHB strains.

**Figure 5 microorganisms-09-01551-f005:**
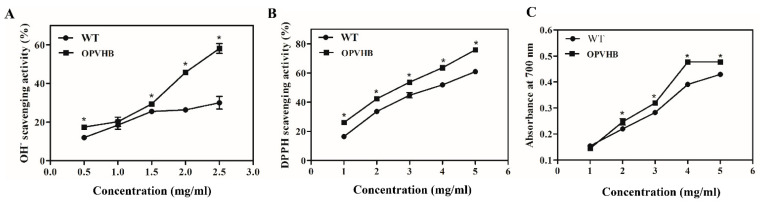
(**A**) OH. radical scavenging activity, (**B**) DPPH radical scavenging activity, and (**C**) reducing power of exopolysaccharides from the WT and OPVHB strains. * indicates significantly different from value for WT (*p* < 0.05).

**Table 1 microorganisms-09-01551-t001:** Monosaccharide composition of exopolysaccharides from the WT and OPVHB strains.

Exopolysaccharides	Monosaccharide Compostion (%)							
	Glc	Gal	Man	Fuc	Rha	Xyl	Ara	GlcUA
WT	36.3 ± 0.5	19.7 ± 0.4	24.9 ± 0.6	4.3 ± 0.5	2.6 ± 0.1	2.8 ± 0.1	2.9 ± 0.1	6.5 ± 0.3
OPVHB	35.4 ± 0.4 *	23.1 ± 0.2 *	31.5 ± 0.5 *	3.2 ± 0.2 *	1.4 ± 0.1 *	0.9 ± 0 *	1.1 ± 0.1 *	3.4 ± 0.4 *

* indicates significantly different from value for WT (*p* < 0.05).

## Data Availability

Data are available from the corresponding author upon reasonable request.

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
