# Peer review of "Effects of Efficient Expression of Vitreoscilla Hemoglobin on Production, Monosaccharide Composition, and Antioxidant Activity of Exopolysaccharides in Ganoderma lucidum"

_microorganisms, 2021, doi:10.3390/microorganisms9081551_

Round 1
Reviewer 1 Report
This is a valuable contribution to the vhb research field. I think it deserves to be published but I have some concerns:
- The present manuscript is an extension of a previously published study. It is important to stress what is new in comparison with the other study.
- How has the codon optimisation been achieved? The supplementary figure should also include the amino acid sequence.
- What is the actual molecular weight of the VHb protein? Does it carry the intron as an N-terminal extension? If so, does this influence the VHb activity or expression level?
- Why is there such a big dfference in the carbohydrate composition between the wt and the VHb strains? Is this significant?
- The study is based solely on observations, it is essential to include some explanations or possible interpretations of the changes seen.
- Figure 1A should give the size of the plasmid.
- I found some typing errors on lines 88, 166, 333 and 394.
Reviewer 2 Report
Line 45, G. lucdium is spelled incorrectly.
Lines 100, 131 and 133, Use the low-case letters (v/v); (w/v).
Line 151, Check the units (8.8 mmol L−1) and revised all units in the same format.
Line 155, Remove the repeat things =(%)
Figure 1C, the quality of the gel image was too poor, it should be replaced with a high-quality picture.
Figures 3 and 5, What are the stars? Describe them.
Do you know the protein/lipid content of the fungal biomass? How may VHb technology effect the protein or carbohydrate contents of the biomass?
